# DNA Dyes—Highly Sensitive Reporters of Cell Quantification: Comparison with Other Cell Quantification Methods

**DOI:** 10.3390/molecules26185515

**Published:** 2021-09-11

**Authors:** Anna Ligasová, Karel Koberna

**Affiliations:** Institute of Molecular and Translational Medicine Dentistry, Faculty of Medicine and Dentistry and Czech Advanced Technology Research Institute, Palacký University in Olomouc, Hněvotínská 5, 779 00 Olomouc, Czech Republic

**Keywords:** DNA dyes, cell quantification, enzymatic conversion of substrate, cell metabolism

## Abstract

Cell quantification is widely used both in basic and applied research. A typical example of its use is drug discovery research. Presently, plenty of methods for cell quantification are available. In this review, the basic techniques used for cell quantification, with a special emphasis on techniques based on fluorescent DNA dyes, are described. The main aim of this review is to guide readers through the possibilities of cell quantification with various methods and to show the strengths and weaknesses of these methods, especially with respect to their sensitivity, accuracy, and length. As these methods are frequently accompanied by an analysis of cell proliferation and cell viability, some of these approaches are also described.

## 1. Introduction

The simplest method of cell quantification is probably the use of a hemocytometer or, in the case of adherent cells, direct calculation of cells using a microscope. Both methods are simple and provide the possibility to quantify live and fixed cells. On the other hand, the method based on a hemocytometer is a very time-consuming process. In this respect, the use of a hemocytometer for the quantification of multi-well plates is practically impossible. Although the development of automatic microscopic stations equipped with a camera and convenient software provides one possibility to evaluate a high number of samples, this solution is relatively expensive. Moreover, as the process of cell quantification using automatic microscopic stations requires the acquisition, processing, and evaluation of a large quantity of data, it is still relatively slow.

On the other hand, automatic microscopy stations are a suitable choice for the evaluation of some aspects of the viability of the cell population. For example, microscopes are typically used to discern dead and live cells with trypan blue staining [1]. Microscopy is also suitable for the analysis of cell proliferation by specific proliferation markers such as PCNA (proliferating cell nuclear antigen), Ki-67 antigen, and/or phospho-histone H3.

Much faster methods of cell quantification are based on the enzymatic conversion of various substrates into detectable products as, typically, plate readers are used for the sample evaluation. Already in 1983, Mosmann [2] developed a colorimetric assay for cell quantification based on the enzymatic conversion of a yellow substrate to a blue product. Although it is a frequently used approach for cell quantification, it can provide non-linear data [3] as higher cell densities provide a lower signal per cell than lower densities. The dependence of the signal on the metabolism of cells is therefore the main drawback of enzymatic methods. Another disadvantage is the difficulty to quantify the fixed cells or the necessity to perform the signal measurement relatively soon after the reaction with the substrate.

The dependence of enzymatic assays on the enzyme activity can be overcome by methods based on DNA quantification as the content of DNA does not depend on cell metabolism. Here, such type of methods is summarized, and a comparison with the frequently used alternative approaches is provided. In addition, the chosen methods of analysis of cell proliferation and cell viability are described.

## 2. Fluorescent DNA Dyes

Fluorescent DNA dyes are substances whose fluorescence is substantially increased when they are bound to DNA [3]. Several of these dyes were tested for cell quantification. Initially, Hoechst dyes, DAPI, or propidium iodide were used. More recently, newly developed dyes such as PicoGreen, CyQuant GR, or SYBR Green I have successfully been applied for cell quantification. 

Hoechst dye 33258 and its derivatives Hoechst 33342 and Hoechst 34580 belong to the bis-benzimide group of fluorescent dyes developed by the German company Hoechst AG in the early 1970s as substances with a potential clinical outcome (Figure 1) [4,5]. They are excited by UV light (~360 nm). When bound to DNA, their fluorescence increases by approximately thirty times. Hoechst dyes are non-intercalating stains which preferentially bind to the A:T-rich areas in the minor groove of the DNA [5,6]. However, the possible intercalation of Hoechst dyes in G:C-rich areas was also shown [6].

As the cell membrane is more permeable for Hoechst 33342 than for the other Hoechst dyes, it is preferentially used for the analysis of live cells, typically for the measurement of DNA content [7,8]. However, one must take into consideration in such experiments that the long incubation period of live cells with Hoechst 33342 can lead to pronounced ATM, Chk2, and p53 phosphorylation [9]. Although Hoechst 33342 can be used to stain live cells, procedures for cell quantification typically involve fixation. For example, the approach described by McCaffrey et al. [10] involves fixation by formaldehyde or glutaraldehyde, staining, washing, and signal measurement [10]. The signal measurement can be conducted by a plate reader, although microscopes can be used as well. As the fixed cells are quantified, samples can be stored for a prolonged period. 

The permeability of cell membranes for Hoechst 33258 is much lower; therefore, the protocols for DNA and cell quantification include cell fixation or cell lysis [11,12,13,14,15]. In the study of Rage et al. [15], the unfixed cells were lysed by incubation in distilled water and frozen. After thawing the samples, a solution of Hoechst 33258 was added, followed by fluorescence measurement. As the samples can be frozen before cell quantification, this protocol provides the possibility to store samples for a long time before cell quantification as well. 

Another protocol based on Hoechst 33258 staining includes cell lysis with sodium dodecyl sulphate (SDS) [11]. It is a multistep protocol involving the removal of adherent cells to the solution, centrifugation, incubation with 1% SDS, and 100-fold dilution of the samples to decrease the SDS concentration prior to the addition of Hoechst [11]. Although no data about the use of this protocol in multi-well plates are available, this protocol seems to be less suitable for such samples.

Another fast protocol is based on methanol fixation followed by staining with Hoechst 33258. The Hoechst dye is then extracted from DNA by denatured ethanol [12]. This protocol can be used for the evaluation of samples containing between 1000 and 500,000 cells [12]. However, as denatured ethanol was used in that study, it is not completely clear if the same result can be obtained with pure ethanol. 

DAPI is another DNA dye frequently used for cell quantification. DAPI belongs to the indol substances (Figure 2a), and initially, it was synthesized by Dann et al. as a potentially active trypanocidal diamine [16,17]. It strongly binds to A:T sequences of the minor groove of DNA and intercalates in G:C or in mixed G:C and A:T sequences of DNA [18]. Similar to the case of Hoechst dyes, after binding of DAPI to DNA, its fluorescence increases (circa twenty times) [19]. DAPI was also successfully used for cell quantification [10]. The protocol included five simple steps: washing, fixation with formaldehyde or glutaraldehyde, staining with DAPI, washing, and fluorescence reading. In this respect, the described method is simple and fast.

Recently, a highly sensitive approach using DNA dyes (Hoechst 33258 or Hoechst 33342, or DAPI) for cell quantification was described [3]. The method overview is shown in Figure 3. The protocol is based on the incubation of fixed cells with a DNA dye, followed by incubation with SDS solution [3]. The presence of SDS results in the quick de-staining of DNA and, simultaneously, in an up to a 1000-fold increase in the fluorescence intensity of the used dyes. This increase can be attributed to the micelle formation of SDS. The sensitivity of this method is around 50–70 cells.

The described approach was also successfully tested for the analysis of the cytotoxic effect of various substances [3]. It, similar to other methods based on DNA staining, does not depend on the metabolic state of the cells. Therefore, it is also suitable for the quantification of cells exhibiting low metabolic activity, including senescent cells.

Propidium iodide is another DNA dye which can be used for cell quantification. This red DNA stain requires cell membrane permeabilization before it can enter the cell, as the intact cell membrane is impermeable for this dye. Propidium iodide belongs to the phenanthridinium group of compounds (Figure 2b) and interacts with double-stranded DNA (dsDNA) and/or RNA, having no preference in nucleosides [20,21]. The fluorescence of propidium iodide increases up to 30-fold if bound to DNA [22]. The sensitivity of the protocol based on the measurement of the propidium iodide signal could be around 150–500 cells. The protocol based on propidium iodide was successfully tested for cytotoxicity assessment, for example, in [23]. However, as cell permeabilization is required, it prolongs the whole procedure. In this study, the cell membrane was permeabilized using freezing at −20 °C for 24 h. Despite the relatively high sensitivity of the protocol, this dye is rather used for the assessment of cell viability, including bacterial cells. In such cases, a combination of propidium iodide with another dye such as SYBR Green I, SYTO 9, or fluorescein diacetate (cell membrane is permeable for these dyes) is commonly used [24,25,26]. On the other hand, it was shown that the protocols based on the use of propidium iodide can lead to an underestimation of viable adherent bacterial cells [25].

PicoGreen belongs to the newer fluorescent DNA dyes with a high specificity for dsDNA (Figure 2c) [21]. The fluorescence of PicoGreen after its binding to DNA is increased circa 1000 times and is proportional to the DNA amount [27]. According to Blaheta et al. [21], circa 100 cells or 0.5 ng DNA can be detected by PicoGreen. As the cell membrane is impermeable for PicoGreen, the cells have to be fixed or digested before incubation with this dye [21]. In this respect, PicoGreen requires around 20 h of digestion of cells with papain to obtain such high sensitivity [21]. Although it was shown that the PicoGreen assay is suitable to measure cell proliferation in 3D cell cultures [28] or to analyze the DNA content in solutions [27,29], Pabbruwe et al. showed that the PicoGreen assay used for DNA quantification in 3D culture is less sensitive than the MTT assay [30]. 

CyQuant GR is a dye available as a part of the commercial kit CyQuant™ Cell Proliferation Assay (ThermoFisher Scientific) that provides a strong green fluorescence when bound to DNA and six times lower fluorescence when bound to RNA [20]. According to [26], the sensitivity limit of cell quantification by this dye is extremely high as it can reveal 10–50 cells. Linearity of the signal was obtained for up to 25,000–50,000 cells. If the concentration of the dye was increased, the linearity was increased up to 100,000–250,000 cells [31]. The disadvantage of the assay based on CyQuant GR is the necessity to freeze and thaw cells at −70 °C followed by cell lysis [3,31]. A variant of the CyQuant kit (CyQuant™ Direct Cell Proliferation Assay, ThermoFisher Scientific) without the need to freeze is available as well, but the sensitivity and the linearity range are lower (100–20,000 cells) [3]. Besides this, it is supposed that the CyQuant assay may interfere with some chemicals present in the culture medium, such as phenol red. 

SYBR Green I is another fluorescent DNA binding dye which was used to measure the cell number by the determination of DNA content [32]. It preferentially binds to dsDNA [33] and has a similar structure to PicoGreen (Figure 2d). SYBR Green I has a low background in the absence of DNA. Therefore, the unbound dye does not need to be removed from samples as it does not fluoresce when not bound to DNA. The sensitivity of the SYBR Green I assay is proposed to be around 1000 cells [32]. This assay was used, for example, to determine the impact of different culture medium compositions on the proliferation of fibroblasts [34].

## 3. Alternative Approaches of Cell Quantification

Alternative methods are mainly based on the measurement of cell metabolic activity. Although methods based on the metabolic activity of cells are simple, fast, and sensitive, the dependence of the signal on the cell number is commonly less linear than in the case of DNA dyes [3] as the metabolic activity can vary greatly with the cell density. Furthermore, they should not be used if substances affecting cellular metabolism are tested [31].

### 3.1. Assays Based on the Measurement of Mitochondrial Activity

#### 3.1.1. Tetrazolium Based Assay

Usually, monotetrazolium salts are used for the measurement of mitochondrial activity. They are reduced by NAD(P)H-dependent oxidoreductases and dehydrogenases by metabolically active cells to formazans [35]. The most commonly used salt is 3-(4,5-dimethyl-2-thiazolyl)-2,5-diphenyl-2H-tetrazolium bromide (MTT, Figure 4). As the reduction of the yellow MTT leads to the formation of insoluble purple formazan crystals (Figure 5), formazan crystals have to be solubilized before the signal measurement. Typically, the signal is measured at 540–720 nm using plate readers [36]. 

Various solutions are used to solubilize formazan crystals. The original colorimetric MTT assay was developed by Mosmann in 1983 [2,35]. He used acid isopropyl alcohol for formazan solubilization [37]. Later, Carmichael et al. showed that the use of acid isopropyl alcohol could lead to low optical absorption values, and therefore they suggested DMSO or mineral oils as an alternative [37,38]. In addition, SDS/HCl solution [39] or ethanol with acetic acid [40] was used as a formazan crystal solvent. Although it is not exactly clear which of the aforementioned solvents is the best one for formazan solubilization, DMSO and isopropanol are the most commonly used solvents for the solubilization of formazan crystals [37,40].

The necessity to solubilize formazan led to the development of alternative monotetrazolium salts. Examples are XTT (sodium 2,3-bis(2-methoxy-4-nitro-5-sulfophenyl)-5-[(phenylamino)carbonyl]-2H-tetrazolium inner salt; Figure 4), MTS (5-[3-(carboxymethoxy)phenyl]-3-(4,5-dimethyl-2-thiazolyl)-2-(4-sulfophenyl)-2H-tetrazolium inner salt; Figure 4), and WST-1 (sodium 5-(2,4-disulfophenyl)-2-(4-iodophenyl)-3-(4-nitrophenyl)-2H-tetrazolium inner salt; Figure 4). The reduction of these modified tetrazolium salts resulted in the formation of water-soluble formazans [36,41,42]. In the case of XTT, the formazan crystals can be directly solubilized in the culture medium [42]. Comparing to MTT, it is supposed that the XTT assay is more sensitive as well [36,41]. MTS needs the presence of an electron-coupling agent such as phenazine methosulphate (PMS) and is similar to XTT reduced to water-soluble formazan in living cells [41]. WST-1 belongs to the new generation of water-soluble tetrazolium salts. It contains iodine in the molecule and is more stable than XTT or MTS [43].

All the tetrazolium-based assays were successfully used for the evaluation of the impact of the tested compounds, drugs, growth factors, or other studied substances on cell viability, comparing to control samples. They were also employed in studies focused on drug resistance [36]. 

#### 3.1.2. Alamar Blue Assay

Besides methods based on the reduction of tetrazolium salts by mitochondrial enzymes, the reduction of resazurin (Alamar Blue) is also frequently used [44]. Resazurin is an oxidized form of 7-hydroxy-3H-phenoxazin-3-1-10-oxide which is converted to the reduced form—resorufin (Figure 6) [45]. The reduction is conducted by mitochondrial enzymes with diaphorase activity [45,46]. The reaction is accompanied by a change of the poorly fluorescent blue resazurin to the highly fluorescent red resorufin [45]. It is supposed that dead or damaged cells produce a lower fluorescence change than proliferating cells [36,47]. 

The Alamar Blue assay was used for an analysis of the impact of ionizing radiation on cell viability and cell re-growth depending on the time and radiation dose [45], cytotoxicity [46,47,48,49], apoptosis and cell death [50,51], neuronal viability [52], and others. It is also suitable for long-term experiments without killing the cells [36,53].

### 3.2. Assays Utilizing Other Cellular Metabolic Activities

There are plenty of other assays employing metabolic activities to quantify cells. Several commonly used methods are described below.

The cellular esterase assay utilizes the production of a fluorescent substance from the non-fluorescent fluorescein esters (fluorescein diacetate or 4-methylumbelliferyl heptanoate) by cellular esterases [54]. The cell membrane is permeable for these esters and impermeable for their fluorescent products. Therefore, incubation with the esters results in the staining of live cells with an intact cell membrane. The sensitivity of the method is circa 100 cells if 4-methylumbelliferyl heptanoate is used [54].

Instead of fluorescein esters, calcein-AM (calcein-acetoxymethylester) can be used. It also passes across the cell membrane of live and dead cells [55,56]. In the cytoplasm, the non-fluorescent calcein-AM is cleaved by the cellular esterases into fluorescent calcein, which is lipid-insoluble, and therefore it remains inside the cells with an intact cell membrane and is released from the dead cells [36,55,56]. The conversion of non-fluorescent calcein-AM is accompanied by the green fluorescent signal of calcein [36,55]. The calcein-AM assay was used, for example, to analyze the cell-mediated cytotoxicity of T lymphocytes or NK cells [55,56]. Microscopes, flow cytometers, or plate readers can be used for the signal detection of both esterase products.

Another approach used for cell quantification is based on the analysis of cytosolic acid phosphatase activity [57,58]. This assay is based on the hydrolysis of *p*-nitrophenol phosphate to *p*-nitrophenol by the intracellular acid phosphatases in viable cells and the measurement of the absorbance of *p*-nitrophenol at 405 nm [58]. The signal is directly proportional to the cell number in the range of 1000–100,000 cells per well of the 96-well plates [58,59,60]. This assay was used to determine the cell number in various cell lines [58,60]. 

The ATP assay can be used for cell quantification as well. ATP is the major energy source produced mainly in mitochondria [36] and is accepted as a marker of viable cells [60]. In this respect, the total ATP levels can be employed to evaluate the cell viability or cytotoxic effect of various substances. The ATP assay utilizes the determination of the ATP level by the conversion of the added luciferin to oxyluciferin by the enzyme luciferase in the presence of Mg^2+^ ions and ATP, producing the luminescent signal in cell lysates [61,62]. This assay is fast, and its sensitivity limit is approximately 1500 cells [61]. The disadvantage of the ATP assay is that it cannot be used in studies with substances affecting cellular metabolism without time-consuming controls [31].

## 4. DNA Synthesis and Proliferation Markers

As the proliferative activity of tumour cells is an important prognostic marker in the diagnosis of cancer and the proliferative activity is also followed during the development of anticancer drugs, proliferation markers are widely used in basic and applied research [63]. Basically, two groups of markers are available: (i) labeled DNA precursors and (ii) cell cycle-dependent proteins. While DNA precursors are primarily used during the development of anticancer drugs and also in studies dealing with the impact of various substances on the cell cycle, cell cycle-dependent proteins are predominantly used in clinics.

DNA precursors mainly involve nucleoside analogues. These analogues can pass the cell membrane and are incorporated by cellular polymerases into DNA. If a short pulse with an analogue is used, most of the labeled cells are in the S phase. If a long pulse is used, only proliferating cells are labeled.

In the past, [^3^H]-thymidine was most frequently used for this purpose. After its incorporation into newly synthesized DNA, the cell population is analyzed by autoradiography or scintillation [28,36]. However, one clear disadvantage is the fact that this assay is time-consuming and requires radioactive compound handling. Therefore, other DNA replication markers were developed. The most popular are 5-bromo-2′-deoxyuridine (BrdU) and 5-ethynyl-2′-deoxyuridine (EdU) [36]. 

At the beginning of its use, BrdU incorporated into DNA was detected by autoradiography, similarly as [^3^H]-thymidine. Later on, specific antibodies against BrdU were developed, and BrdU became very popular in analyses of DNA synthetic activity [64]. On the other hand, the disadvantage of BrdU is the necessity to reveal it in the DNA structure to be accessible for antibodies. Presently, various methods of BrdU revelation are available. These include the use of acids [65] or hydroxide [66], the use of nucleases [67], the use of low-concentration HCl with exonuclease III [68], incubation with copper ions [69], photolysis [70], or heat denaturation [66]. 

EdU works on a similar principle as BrdU as it is incorporated into DNA during DNA replication instead of thymidine [71]. Nevertheless, the incorporated EdU is detected by the so-called click reaction and does not need to be revealed in the DNA structure [72]. On the other hand, EdU is not suitable for long pulses as it is toxic for cells [64]. The quantification of BrdU- or EdU-labeled cells is usually performed by flow cytometry or microscopy [67,68]. Microscopy is used for the detection of DNA synthetic activity by BrdU or EdU as DNA replication serves as an excellent proliferation marker [64].

From the protein proliferation markers, PCNA, Ki-67 antigen, or phospho-histone H3 are usually used to assess cell proliferation. PCNA is a protein participating in DNA replication, DNA repair, chromatin remodeling, and cell cycle control [73] and is usually synthesized from the late G1 phase through to the S phase until the G2 phase and is absent in the M phase [74]. PCNA exhibits two patterns of nuclear expression: (1) free PCNA with diffuse nuclear staining expressed in all cells and (2) bound PCNA which is a component of the replication complex showing a punctate pattern only in the S phase [75]. Ki-67 staining is widely used as a proliferation indicator in the clinical setting, although its function and dynamics are not clear [76]. It was initially considered as a marker of cycling cells absent in non-dividing cells [77]. More recent studies showed that it is rather a graded than a binary marker of proliferation [76]. Similarly, phospho-histone H3 is used in the clinical setting. The phosphorylation of histone H3 occurs during the late G2 phase to the early prophase, while dephosphorylation occurs slowly from the late anaphase to the early telophase. This allows using phospho-histone H3 staining for the detection of mitotically active cells [78]. 

## 5. Viability Test

Dye exclusion assays are frequently used to estimate the number of viable cells versus dead cells in tested cell populations. They are based on the exclusion of the used dye by live cells due to the intact cell membrane. On the contrary, dead cells do not exclude the dye and are stained [1,36]. However, although cells may have an intact cell membrane, they might not grow or proliferate [79]. Therefore, this should be taken into account when using the dye exclusion assay. 

The most commonly used dye in these assays is trypan blue, but other dyes such as eosin, propidium, nigrosin, safranin, methylene blue, and Congo red can be used as well [1,36,80,81,82]. Trypan blue is a negatively charged diazo dye and can be used both in vitro and in vivo. It stains dead cells in blue, while live cells are clear and unstained [36]. 

The evaluation of the dye exclusion assay can be performed either manually using a hemocytometer (such as a Bürker chamber, a Fuchs–Rosenthal chamber, a Nageotte chamber, a Malassez chamber, a Thoma chamber, or a Bürker–Türk chamber [83]; Neubauer chamber, Figure 7), semi-automatically (e.g., Countess 3 automated cell counter from Thermo Scientific), or automatically (e.g., Vi-CELL XR from Beckman Coulter) [80,81]. Eventually, a flow cytometer or automatic microscopy station with the appropriate software can be used. 

The direct cell counting by a hemocytometer is very accurate; however, the accuracy rapidly decreases for a small number of cells [12]. Furthermore, this approach is time-consuming and laborious [12,31]. 

The quantification of necrotic and/or apoptotic cells can also be conducted by the lactate dehydrogenase release assay (LDH; Figure 8). This assay was developed to study cytotoxicity in immune cells, but presently, it is also used to study the impact of new compounds or drugs [36,84,85]. LDH is released to the culture medium after membrane disruption in dead cells. LDH catalyzes the conversion of lactate to pyruvate and the simultaneous conversion of NAD^+^ to NADH in the first step [84,85,86]. The produced NADH is then used in the following step to reduce the tetrazolium salt into formazan which is measured colorimetrically. The measured signal of formazan is directly proportional to the LDH released to the culture medium [36]. 

Another possibility for the quantification of necrotic and/or apoptotic cells is to use the glucose-6-phosphate dehydrogenase release assay (G6PD; Figure 8) [87]. This enzyme is also released from cells with a damaged cell membrane. It generates NADPH from NADP^+^ which is, in the next step, used for the reduction of the exogenously added resazurin to fluorescent resorufin [36,87]. The principle of both release assays is shown in Figure 8. 

Both the LDH and G6PD assays are very fast as they can be completed in around 1 h [87,88]. On the other hand, the LDH release assay seems to be less sensitive than the G6PD release assay. It also provides higher background signals in most serum-containing growth media [36,87]. Despite that, the LDH release assay is a widely used and accepted assay for cell viability analysis [36,89].

The alternative approach utilizes the determination of glyceraldehyde-3-phosphate dehydrogenase activity in dead cells. It is performed in two consecutive steps. In the first step, the enzyme glyceraldehyde 3-phosphate dehydrogenase, which is abundant in the cytoplasm, converts the added glyceraldehyde 3-phosphate into 1,3 diphosphoglycerate. In the consecutive step, the added phosphoglycerate kinase catalyzes the transfer of the phosphate group from 1,3 diphosphoglycerate to adenosine diphosphate. The result of this reaction is ATP and 3-phosphoglycerate. ATP is then used as a substrate for luciferase. The advantages are the relative sensitivity, versatility, and, to some degree, independence from the cell type [90]. However, it was also found that this enzyme release assay can be inaccurate in measuring cytotoxicity in a heterogeneous mixture of effector and target cells; therefore, other assays utilizing enzyme activities were developed [36,91].

## 6. Conclusions and Method Overview

Various assays have been developed for cell quantification. For the sake of simplicity, we summarize the approaches described in this review with respect to the principle, the method of their evaluation, advantages/disadvantages, and sensitivity in Table 1.

## 7. Patents

Palacký University Olomouc holds Czech patents (307415, 308519, 308385) and a European patent (3395956) for the method of determining the amount of DNA in samples and its use for determining the amount of cells, for the method of stabilizing cells and for selectively removing DNA dyes. Names of inventors: A.L. and K.K.

## Figures and Tables

**Figure 1 molecules-26-05515-f001:**
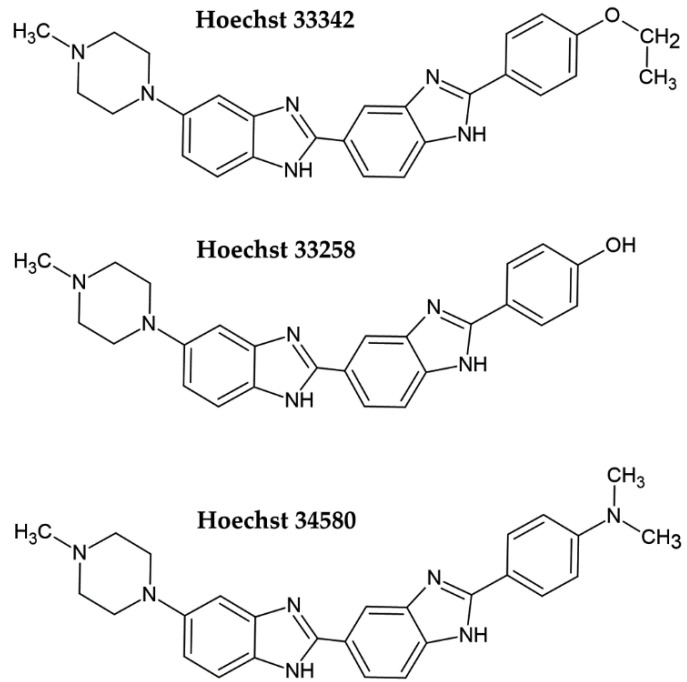
Chemical formulae of the commonly used Hoechst dyes.

**Figure 2 molecules-26-05515-f002:**
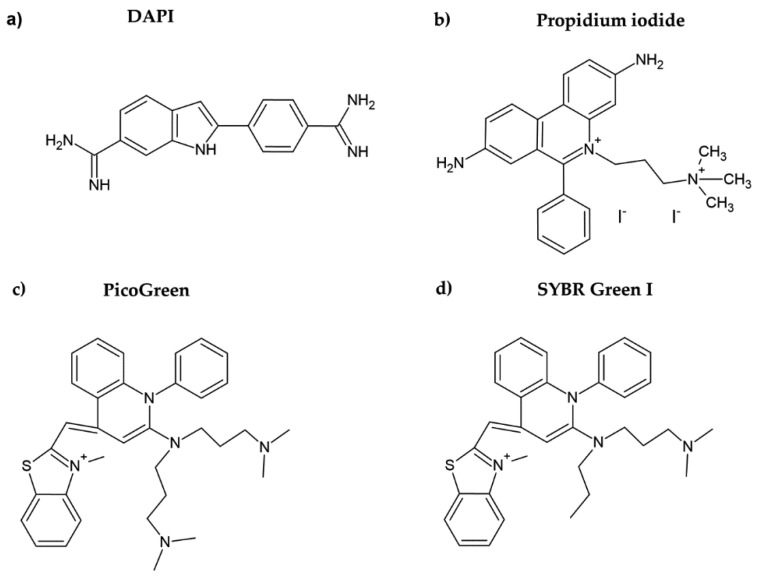
Chemical formulae of DAPI (**a**), propidium iodide (**b**), PicoGreen (**c**), and SYBR Green I (**d**).

**Figure 3 molecules-26-05515-f003:**
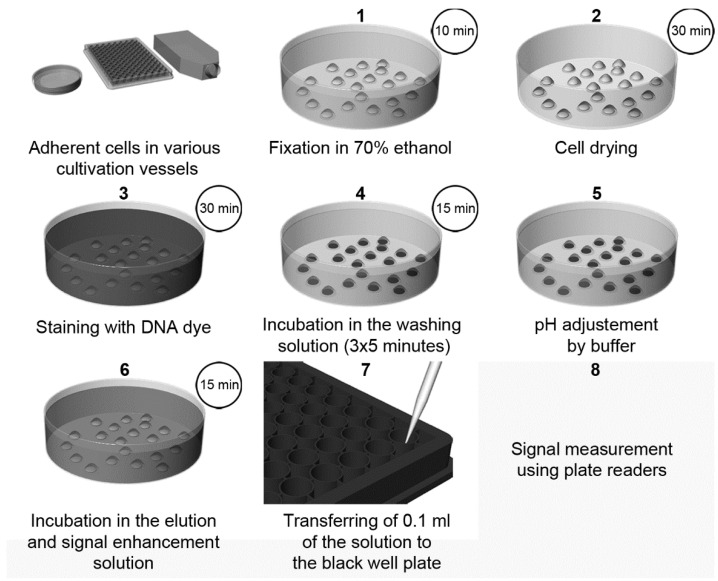
The scheme of cell quantification according to [3]. The cells are fixed with ethanol followed by drying. Then, cells are stained with Hoechst dye. Afterwards, the cells are washed for 3 × 5 min to remove the non-specifically bound dye and to stabilize the cells. After a short rinsing of cells with a buffer, adjusting the pH to 7, samples are incubated in the elution and signal enhancement solution with SDS. Finally, small aliquots are transferred to black well plates, and the signal is measured and evaluated. Step 2 (cell drying) is optional, but it is highly recommended if 96-well plates are used. If DAPI is used instead of Hoechst dyes, the procedure and incubation times are the same as in the case of Hoechst dyes except for the washing in Step 4, which should be shortened to only 3 × 2 min. Adapted from [3].

**Figure 4 molecules-26-05515-f004:**
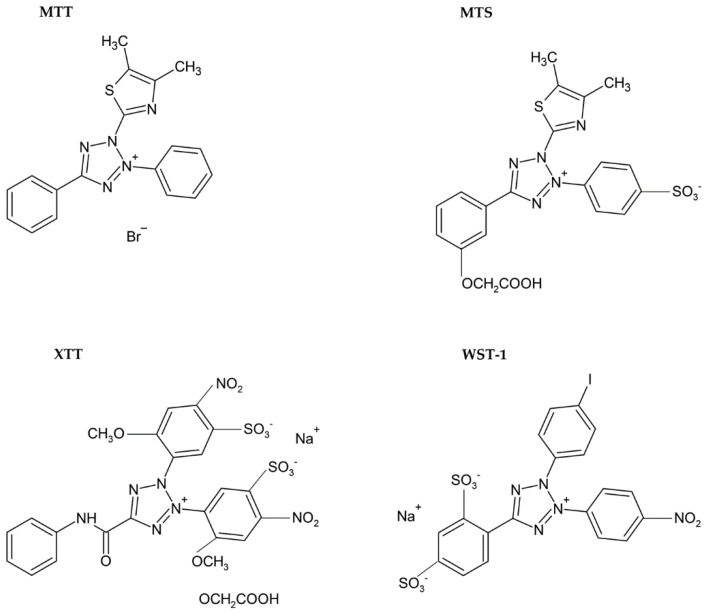
Chemical formulae of tetrazolium salts.

**Figure 5 molecules-26-05515-f005:**
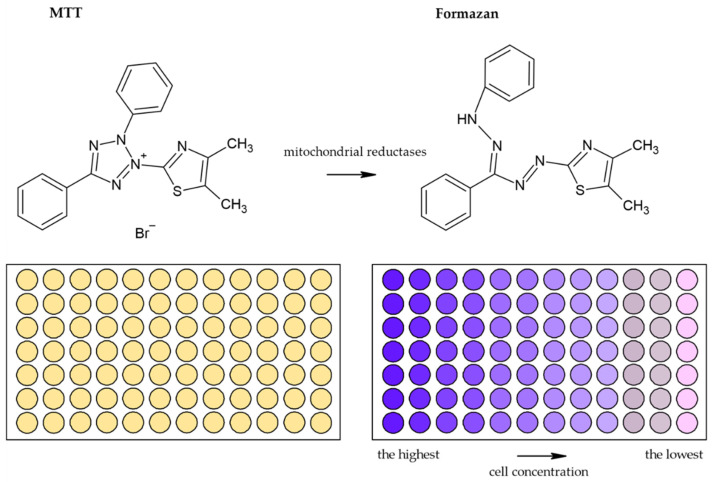
Scheme of the reduction of MTT to formazan. Scheme of the reduction of MTT to formazan including the color change is shown. Similarly, such conversion and color change are the principle in the XTT, MTS, and WS-1 assays.

**Figure 6 molecules-26-05515-f006:**
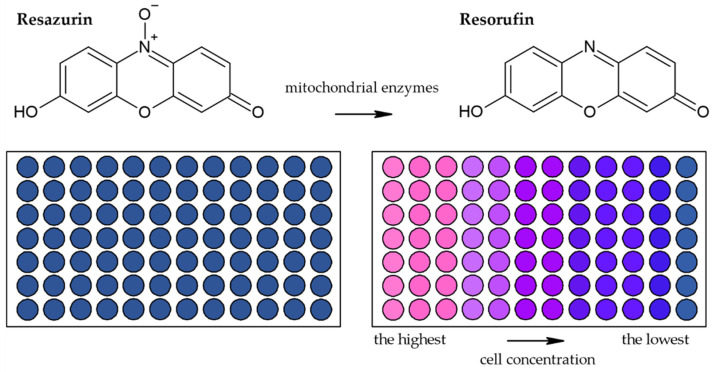
Scheme of the reduction of resazurin to resorufin including the color change.

**Figure 7 molecules-26-05515-f007:**
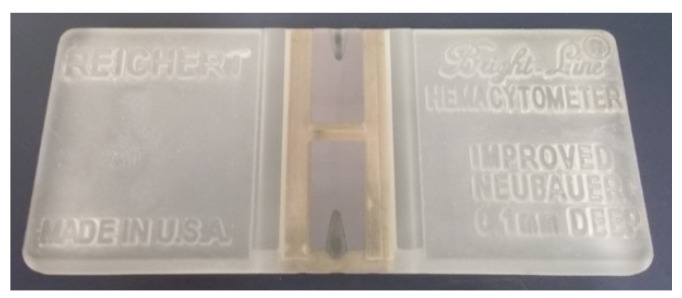
An example of a hemocytometer (improved Neubauer counting chamber). For other types of hemocytometers, see, e.g., [83].

**Figure 8 molecules-26-05515-f008:**
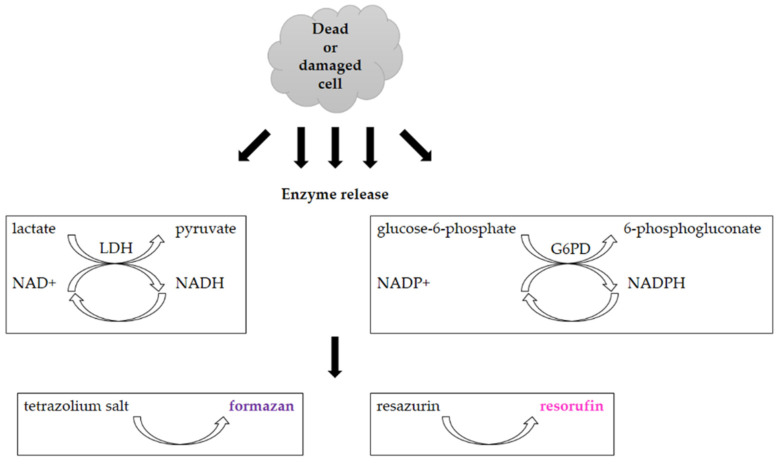
Scheme of the principle of the lactate dehydrogenase release assay and glucose-6-phosphate dehydrogenase release assay.

**Table 1 molecules-26-05515-t001:** Summary of the discussed methods used for cell quantification. The light red-colored rows depict approaches using DNA dyes; the light blue-colored row depicts methods based on the analysis of DNA replication; and the light green-colored rows depict methods based on the measurement of metabolic activity.

Method	Method of Evaluation	Advantages	Disadvantages	Sensitivity
Increase in the fluorescence of DAPI, Hoechst 33258, or Hoechst 33342 after their binding to DNA in the fixed cells [10,14]	Fluorescence (plate reader)	Cheap and fast Samples can be stored for a prolonged time	The necessity of fixation step	N/A
Elution of Hoechst 33258 from DNA of fixed cells by denatured ethanol [12]	Fluorescence (plate reader)	Cheap and fast	The necessity of fixation step Unclear whether non-denatured ethanol can be used	1000 cells
Increase in the fluorescence of Hoechst 33258 after its binding to DNA of lysed cells [15]	Fluorescence (plate reader)	Cheap and fast Samples can be stored in the freezer before the signal measurement	Cell lysis is necessary	N/A
Increase in the fluorescence of Hoechst 33258 after its binding to DNA of lysed cells [11]	Fluorescence (plate reader)	Cheap	Multistep procedure Cell lysis is necessary Less suitable for well plates	N/A
Elution of Hoechst or DAPI dyes from the DNA of the fixed cells and signal enhancement by SDS [3]	Fluorescence (plate reader or microscope)	Cheap, fast, and simple High-throughput performance possible Signal stable for at least twenty days No cell lysis Possible combination with the detection of cellular components. Suitable for slowly growing or senescent cells	The washing step has to be controlled in the case of DAPI Necessity of fixation step	70 cells (DAPI) 35 (Hoechst 33342)
Increase in the fluorescence of propidium iodide after binding to DNA of permeabilized cells [23]	Fluorescence (plate reader)	Cheap Sensitive Cell viability can be simultaneously evaluated	Time-consuming	150–500 cells
Increase in the fluorescence of PicoGreen after its binding to DNA of the enzymatically digested cells [21,28]	Fluorescence (plate reader)	Sensitive Suitable for slowly growing cells	Time-consuming cell digestion necessary Expensive	100 cells
Increase in the fluorescence of CyQuant GR after its binding to DNA of the lysed cells [31]	Fluorescence (plate reader)	Sensitive	Freezing/thawing cycles at −70 °C Cell lysis necessary Expensive	10–50 cells
Increase in the fluorescence of SYBR Green I after its binding to DNA of the fixed and permeabilized cells [24,32]	Fluorescence (phosphorimager)	Low background in the absence of DNA—no need to wash out the unbound dye	Less sensitive	1000 cells
Incorporation of ^3^H-thymidine in replicated DNA [20,28,54]	Autoradiography (β-scintillation counter)	High sensitivity	High costs Work with radioactive material—need to abide by rules Time-consuming Accuracy in high-density cell populations depends on the diffusion efficacy of ^3^H-thymidine	100 cells
Reduction of tetrazolium salts (MTT, MTS, XTT, WST-1) by mitochondrial enzymes to the colored products [2,12,36,92]	Colorimetric (plate reader)	Easy Sensitive Cheap	Relies on the metabolic activity of cells or intracellular enzyme concentration Not suitable for long-term studies (toxic for cells)	200 cells
Reduction of Alamar Blue by the mitochondrial enzymes to the fluorescent product [28,36,44,45,46]	Fluorescence (plate reader, microscope)	Simple Based on water-soluble compound Suitable for adherent and suspension cells Non-toxic Cheap Sensitive Stable in culture medium Suitable for long-term studies	Relies on the metabolic state of cells Accuracy in high-density cell populations depends on the diffusion efficacy of the dye Long optimization Fluorescence interference	80 cells
Production of the fluorescent substance from the non-fluorescent fluorescein esters or calcein-AM by cellular esterases [36,54,56,93]	Fluorescence (plate reader)	Fast Simple	Depends on the metabolic state of cells Less sensitive in adherent cells	100 cells (if 4-methylumbelliferyl heptanoate is used as a substrate)
Conversion of the added luciferin to oxyluciferin by the enzyme luciferase in the presence of ATP and production of luminescence [36,61]	Luminescence	Fast	Cell lysis necessary Cannot be used in studies with substances affecting cellular metabolism	1500 cells
Hydrolysis of *p*-nitrophenol phosphate to *p*-nitrophenol by the intracellular acid phosphatases [57,58,59]	Absorbance (plate reader)	Simple Cheap Adherent and suspension cells	Less sensitive	100–1000 cells

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
