# Peer review of "DNA Dyes—Highly Sensitive Reporters of Cell Quantification: Comparison with Other Cell Quantification Methods"

_molecules, 2021, doi:10.3390/molecules26185515_

Round 1

Reviewer 1 Report

Ligasová and Koberna comprehensively reviewed and compared different cell-quantification DNA dyes. Additionally, the authors discussed some methods used for cell proliferation and cell viability. Only some minor points are suggested.

  1. Legends in Figure3 are not very clear. It seems that it is a general schema for different fluorescent DNA dyes. The legends and Figure3 can be improved to show what dyes fit for this schema and what specific steps are for different dyes. Also, Figure3 was not mentioned in the main text.
  2. Figure7 only shows the photo of the Neubauer chamber. Authors may consider showing other mentioned hemocytometers as well.
  3. From line 334 to line 347, the authors discussed two different enzymes release assays to quantify dead cells. Authors may consider adding one more paraph to discuss what conditions researchers should choose LDH and what conditions should choose G6PD.

Author Response

Thank you for all your valuable comments. Below find our replies.

  1. Legends in Figure3 are not very clear. It seems that it is a general schema for different fluorescent DNA dyes. The legends and Figure3 can be improved to show what dyes fit for this schema and what specific steps are for different dyes. Also, Figure3 was not mentioned in the main text.

We rewrote the legend to Figure 3 (page 4, lines 111-124). We hope that now it is clear. The reference to Figure 3 is on the page 3, lines 105-106.

  1. Figure7 only shows the photo of the Neubauer chamber. Authors may consider showing other mentioned hemocytometers as well.

We rewrote the legend to Figure 7 and added the reference containing images of other types of hemocytometer (page 10, lines 338-339). 

  1. From line 334 to line 347, the authors discussed two different enzymes release assays to quantify dead cells. Authors may consider adding one more paragraph to discuss what conditions researchers should choose LDH and what conditions should choose G6PD.

We added one paragraph enabling more direct comparison of these approaches (page 10, lines 358-362).

Reviewer 2 Report

The authors present a well-written and comprehensive overview of cell quantification methods with special focus on cell quantification using DNA dyes, an area of their own expertise and topic of their recent published papers (specifically ref 3). This review is a nice overview of various methods and will be helpful for many readers.

However, I personally feel that the title is a bit misleading, as only roughly one third of the review focuses on DNA dyes. I would suggest a slightly different title to indicate that the review compares classical cell quantification methods with DNA dyes. 

Overall, the review reads very well and is well annotated. Besides, the suggestion with the title and the few specific issues listed below, I have no further concerns.

Specific pints to be addressed:

Page 1 line 41: maybe replace the term “impossibility” with “difficulties” as technically the cells could still be counted.

Page 3 line 87: wording need to be changed

Page 4 line 118: grammer issues

Figure 2 and 4: formulae instead of formula or formulas

Figure 3: The first line “ The scheme of the method” should be rephrased.

Figure 6: “ctrl” is not shown in the actual figure but mentioned in the legend.

Table 1: title: maybe change the title to: Summary of discussed methods used for cell quantification. I would also suggest to color code the various sections by method (DNA, metabolic, marker, etc) to make the table even more comprehensive. Some columns are not aligned properly – please check.

Author Response

Thank you for all your valuable comments. Below find our replies.

  1. However, I personally feel that the title is a bit misleading, as only roughly one third of the review focuses on DNA dyes. I would suggest a slightly different title to indicate that the review compares classical cell quantification methods with DNA dyes.

We changed the title of the review.

  1. Page 1 line 41: maybe replace the term “impossibility” with “difficulties” as technically the cells could still be counted.

We rewrote the sentence accordingly.

  1. Page 3 line 87: wording need to be changed

We changed the wording.

  1. Page 4 line 118: grammar issues

We rewrote the sentence.

  1. Figure 2 and 4: formulae instead of formula or formulas

We replaced the words “formula and formulas” with the word “formulae”.

  1. Figure 3: The first line “ The scheme of the method” should be rephrased.

We rephrased the mentioned sentence.

  1. Figure 6: “ctrl” is not shown in the actual figure but mentioned in the legend.

We removed the note about „ctrl“ from the figure legend.

  1. Table 1: title: maybe change the title to: Summary of discussed methods used for cell quantification. I would also suggest to color code the various sections by method (DNA, metabolic, marker, etc) to make the table even more comprehensive. Some columns are not aligned properly – please check.

We changed the title of the Table 1 legend. We also divided the table into three categories based on the method principle using different colouring of the table rows. We checked the columns alignment.